# Path Planning of a Mobile Robot for a Dynamic Indoor Environment Based on an SAC-LSTM Algorithm

**DOI:** 10.3390/s23249802

**Published:** 2023-12-13

**Authors:** Yongchao Zhang, Pengzhan Chen

**Affiliations:** School of Intelligent Manufacturing, Taizhou University, Taizhou 318000, China; zycac@126.com

**Keywords:** mobile robot, path planning, SAC-LSTM algorithm, burn-in mechanism, prioritized experience replay mechanism

## Abstract

This paper proposes an improved Soft Actor–Critic Long Short-Term Memory (SAC-LSTM) algorithm for fast path planning of mobile robots in dynamic environments. To achieve continuous motion and better decision making by incorporating historical and current states, a long short-term memory network (LSTM) with memory was integrated into the SAC algorithm. To mitigate the memory depreciation issue caused by resetting the LSTM’s hidden states to zero during training, a burn-in training method was adopted to boost the performance. Moreover, a prioritized experience replay mechanism was implemented to enhance sampling efficiency and speed up convergence. Based on the SAC-LSTM framework, a motion model for the Turtlebot3 mobile robot was established by designing the state space, action space, reward function, and overall planning process. Three simulation experiments were conducted in obstacle-free, static obstacle, and dynamic obstacle environments using the ROS platform and Gazebo9 software. The results were compared with the SAC algorithm. In all scenarios, the SAC-LSTM algorithm demonstrated a faster convergence rate and a higher path planning success rate, registering a significant 10.5 percentage point improvement in the success rate of reaching the target point in the dynamic obstacle environment. Additionally, the time taken for path planning was shorter, and the planned paths were more concise.

## 1. Introduction

Mobile robot path planning is a crucial technique that enables robots to navigate through an environment while avoiding obstacles. This is achieved by planning a collision-free path [1] from the robot’s current position to a desired destination based on environmental information obtained through sensors. Traditional path planning methods encompass a variety of algorithms with different principles and application scenarios. These methods can be classified into global and local path planning algorithms, including graph search methods, random sampling methods, bionic algorithms, artificial potential field methods, simulated annealing algorithms, neural network methods, and dynamic window methods.

Classical graph search algorithms include Dijkstra [2], A* [3], and D* [4]. The Dijkstra algorithm solves the shortest path problem from a single point to all other vertices in a directed graph using a greedy algorithm. Although it can obtain the optimal path, it has a high computational cost and low efficiency. The A* algorithm, based on the Dijkstra algorithm, improves efficiency by adding the heuristic information, but its effectiveness in complex environments is not guaranteed. Moreover, this algorithm is only suitable for static environments and it performs poorly in dynamic environments. The D* algorithm is an improved version of the A* algorithm that can be applied in dynamically changing scenarios.

Classical random sampling methods such as Probabilistic Roadmap (PRM) [5] and Rapidly Exploring Random Tree (RRT) [6] have been widely used in robot path planning and motion control fields. The Lazy PRM [7] algorithm is an improved version of the PRM algorithm that enhances efficiency by reducing the number of calls to the local planner. Liu et al. [8] improved the RRT algorithm by using a goal-biased sampling strategy to determine the nodes and introduced an event-triggered step length extension based on the hyperbolic tangent function to improve node generation efficiency. Euclidean distance and angle constraints were used in the cost function of node connection optimization. Finally, the path was optimized further using path pruning and Bezier curve smoothing methods, leading to an improved convergence and accuracy.

Bionic algorithms are heuristic optimization algorithms based on the evolution and behavior of biological organisms in nature, mainly including genetic algorithms [9] and ant colony algorithms [10]. Liang et al. [11] integrated the ant colony algorithm and genetic algorithm to propose a hybrid path planning algorithm, which used a genetic algorithm to generate initial paths and then used an ant colony algorithm to optimize them, significantly improving the accuracy and efficiency of path planning.

The artificial potential field method [12] was first applied in the mobile robot path planning field in 1986. Zha et al. [13] improved the artificial potential field method by adding a distance factor between the target point and the vehicle in the repulsive force function, and a safety distance within the influence range of obstacles. The experimental results showed that when the vehicle was driving within the safety distance of obstacles, it would be subject to increased repulsive force, ensuring the safety of the vehicle during driving. Zhao et al. [14] proposed a multi-robot path planning method based on an improved artificial potential field and a fuzzy inference system, which overcame the problem of smooth path planning existing in traditional artificial potential field methods by using the incremental potential field calculation method.

Afifi et al. [15] proposed a vehicle path planning method based on a simulated annealing algorithm, which solved the vehicle path planning problem under time window constraints, using the simulated annealing algorithm for path optimization and search. Jun et al. [16] proposed a particle swarm optimization combined with simulated annealing (PSO-ICSA) to self-adaptively adjust the coefficients, enabling high-dimension objects to enhance the global convergence ability.

Zhang et al. [17] proposed an indoor mobile robot path planning method based on deep learning. They used deep learning models to extract features from sensor data to predict the robot’s motion direction and speed, achieving good results in experiments. During robot path planning, when the environment changes, the robot re-executes the algorithm, significantly increasing the time to find the optimal path. Dimensionality reduction seems to be a solution to this problem. Ferreira et al. [18] proposed a path planning algorithm based on a deep learning encoder model. They built a CNN encoder that uses nonlinear correlations to reduce data dimensions, eliminating unnecessary information and accelerating the efficiency of finding the shortest path.

Bai et al. [19] combined a dynamic window algorithm with the A* algorithm to propose an unmanned aerial vehicle path planning method. Experiments proved that the improved algorithm significantly reduced both path planning length and time. Lee et al. [20] proposed a dynamic window approach based on finite distribution estimation. Experimental results showed that this method could achieve a reliable obstacle avoidance effect for mobile robots and exhibited good performance in multiple scenarios.

Most of the above methods heavily rely on environmental map information. When faced with unknown environments, these methods may not achieve ideal results. When mobile robots are in unknown environments, due to the lack of environmental cognition, they must have certain exploration and autonomous learning abilities to efficiently complete path planning tasks. Therefore, studying mobile robot path planning algorithms that rely on little or no map information of the environment and have autonomous learning abilities has become one of the current key research topics.

Recently, artificial intelligence technologies represented by deep learning and reinforcement learning have developed rapidly. Reinforcement learning algorithms do not rely on map information and can learn path planning strategies in unknown environments by interacting with the environment through trial and error. However, reinforcement learning is prone to the problem of dimensionality disaster in the path planning process. Deep learning is an end-to-end model that can fit the mapping relationship between high-dimensional input and output data, and is suitable for dealing with high-dimensional data problems. Deep reinforcement learning combines the advantages of deep learning and reinforcement learning and has a huge advantage over other path planning methods when dealing with complex unknown environments. Nevertheless, path planning methods based on deep reinforcement learning still face problems such as sparse rewards, a slow learning rate, and difficult convergence in application process.

This paper focuses on the indoor mobile robot path planning problem in complex and unknown dynamic environments with several static and dynamic obstacles, using the Soft Actor–Critic (SAC) algorithm as the main method. The SAC algorithm, while powerful, exhibits certain limitations, namely (1) difficulty in processing complex or dynamic environmental information; (2) inadequacy in handling long-term dependencies in path planning; and (3) lack of predictive capability for future environmental states. To address these shortcomings, an improved SAC-LSTM algorithm is proposed. The main contributions of this paper are as follows. First, the LSTM network with memory capability is introduced into the SAC algorithm, allowing the agent to make more reasonable decisions by combining historical and current states and predict the dynamic changes in the environment, such as the future positions of moving obstacles. Second, the burn-in training mechanism is introduced to solve the problem of memory impairment caused by the hidden state being zeroed during the training process of the LSTM network, stabilizing the learning process, especially in the early stages of training. Third, by combining the prioritized experience replay mechanism, the problem of low sampling efficiency of the algorithm is solved, and the convergence speed of the algorithm is accelerated. Fourth, a complex dynamic test scenario is constructed for indoor mobile robots, featuring multiple stationary and moving obstacles of various sizes and shapes, as well as different motion trajectories, making the test scenario more realistic.

The rest of this paper is organized as follows: Section 2 provides an overview of related works regarding path planning methods in dynamic environments. Section 3 describes the proposed SAC-LSTM system framework and algorithms in detail. Section 4 presents the experimental results and performance analysis. Finally, Section 5 concludes the paper and discusses future research directions.

## 2. Related Work

In recent years, researchers have started to apply deep reinforcement learning (DRL) algorithms to the field of path planning to solve complex problems. In 2016, Tai et al. [21] first applied the DQN algorithm to indoor mobile robots, which could complete path planning tasks in indoor scenarios, but the algorithm had low generalization. Wang et al. [22] introduced an improved DQN algorithm combined with artificial potential field methods to design reward functions, improving the efficiency of mobile robot path planning. However, it could not achieve continuous action output for robots.

Lei et al. [23] proposed a path planning algorithm using a DDQN framework with environment information obtained through LiDAR. It designed a new reward function to address the instability issue during training, improving algorithm stability, but its application was limited to simple scenarios with no guarantee of efficiency in complex situations. Tai et al. [24] used an asynchronous deterministic policy gradient algorithm to build a mapless path planner with input from the mobile robot’s LiDAR-scanned environment information. After a period of training, the mobile robot could successfully reach the designated target, but the planned path was relatively tortuous.

In [25], a deep reinforcement learning-based online path planning algorithm was proposed, successfully achieving path planning for drones in dynamic environments. Zhang et al. [26] combined the advantages of DRL and interactive RL algorithms and proposed a deep interactive reinforcement method for autonomous underwater vehicle (AUV) path tracking, achieving path tracking in a Gazebo-simulated environment. In [27], the DDPG algorithm was extended to a parallel deep deterministic policy gradient algorithm (PDDPG) and applied to multi-robot mapless collaborative navigation tasks.

Wang et al. [28] proposed an end-to-end modular DRL architecture that decomposed navigation tasks in complex dynamic environments into local obstacle avoidance and global navigation subtasks, using DQN and dual-stream DQN algorithms to solve them, respectively. Experiments demonstrated that this modular architecture could efficiently complete navigation tasks. Gao et al. [29] introduced an incremental training mode to address the low training efficiency in DRL path planning. In [30], incorporating a curiosity mechanism into the A3C algorithm provided an additional reward for the exploration behavior of mobile robots, addressing the reward sparsity issue to some extent.

De Jesus et al. [31] proposed a mobile robot path planning algorithm based on SAC, achieving path planning in different scenarios built on ROS. However, the algorithm still faced the problem of sparse environmental rewards. Park, K.-W. et al. [32] employed the SAC algorithm to solve the path planning problem for multi-arm manipulators to avoid fixed and moving obstacles, and used the LSTM network to predict the position of the moving obstacles. The simulation and experimental results showed the optimal path and good prediction of obstacle position. Although the simulation and experimental scenario considered moving obstacles, there was only one moving obstacle, lacking verification for multiple obstacles. Additionally, the multi-arm manipulators used a camera in conjunction with the OpenCV vision algorithm to detect obstacles, which has limited effectiveness in detecting and predicting multiple moving obstacles.

In addition to LSTM, the metaheuristic-based recurrent neural network (RNN) [33] has also been applied to control mobile robotic systems. The metaheuristic-based RNNs often integrate optimization algorithms (such as genetic algorithms and particle swarm optimization) with neural network characteristics. The Beetle Antennae Olfactory Recurrent Neural Network (BAORNN) [34] is a metaheuristic-based control framework used for simultaneous tracking control and obstacle avoidance in redundant manipulators. A key feature of this framework is that it unifies tracking control and obstacle avoidance into a single constrained optimization problem, actively rewarding the optimizer to avoid obstacles by introducing a penalty term in the objective function. The distance calculation is based on the Gilbert–Johnson–Keerthi algorithm, which calculates the distance between the manipulator and obstacles by directly using their three-dimensional geometric shapes. In contrast, the RNN may offer more flexibility in solving specific control and optimization problems, but might not match the LSTM’s proficiency in handling complex sequential data and long-term dependencies.

As seen from the above literature, DRL-based path planning methods have the advantages of not relying on map information and autonomous learning capabilities, making them highly suitable for path planning tasks in unknown environments. However, during application, there are still issues such as sparse rewards, slow learning rates, and difficulty in converging the algorithm. In particular, in dynamic or complex environments, the SAC algorithm may require more accurate environmental models for effective learning. This could be difficult to achieve in practice, particularly in environments with high uncertainty or rapid changes. The SAC algorithm might be insufficient in effectively dealing with highly dynamic or non-stationary environments, where environmental states and dynamics can change rapidly.

## 3. Path Planning System

### 3.1. Framework of Path Planning

The task of the mobile robot path planning algorithm is to plan an optimal path from the starting point to the destination, while minimizing the robot’s movement cost, avoiding obstacles, and enabling the robot to reach the destination as quickly as possible. The deep reinforcement learning algorithm is designed to learn a strategy that enables the agent to maximize its reward through interactions with the environment. Applying this algorithm to the mobile robot path planning task essentially transforms the path planning problem into a reinforcement learning problem. The strategy that allows the robot to quickly reach the target point is learned by having the mobile robot try different actions and interact with the environment. The process of interacting with the environment is a Markov Decision Process (MDP), which requires defining the state space, action space, and reward function. This article proposes the application of the SAC-LSTM algorithm to the mobile robot path planning task, as depicted in Figure 1. In this framework, the SAC-LSTM algorithm combines historical and current states to select an action and controls the mobile robot to execute the action in the environment. After interacting with the environment, the mobile robot receives a reward value and a state value. During the training process, important experience samples are prioritized using the prioritized experience replay, and the algorithm is trained using the burn-in mechanism to obtain an optimal strategy for quickly reaching the target point.

### 3.2. SAC-LSTM Algorithm

The Soft Actor–Critic (SAC) algorithm is an off-policy Actor–Critic algorithm based on the maximum entropy model framework. Off-policy learning improves sample efficiency and reduces training time by utilizing data generated from a behavioral policy to train the target policy. The Actor–Critic framework allows the algorithm to be applied to continuous state and action spaces, thus expanding the range of actions and states that can be selected. To prevent premature convergence and avoid local optima, the SAC algorithm uses a stochastic policy. Additionally, introducing entropy during the learning process improves the algorithm’s ability to explore the environment and enhances its performance and robustness.

#### 3.2.1. Maximum Entropy Principle

Entropy is a measure of uncertainty or randomness in a system, used in information theory. It is inversely proportional to system certainty and directly proportional to system randomness. Let *x* be a random variable with probability density function P(X). The entropy H(P) of X can be calculated according to
(1)$$H\left(P\right)=\underset{X\sim P}{E}\left[-\log P\left(X\right)\right].$$

Classical reinforcement learning algorithms adopt a learning strategy that seeks to maximize the expected value of cumulative rewards. This objective is expressed in the following expression
(2)$$\pi^{*}=\arg\underset{\pi }{\max}Q\left(s,\;a\right)=\arg\underset{\pi }{\max}E\left[\sum_{t=0}^{\infty }R\left(s_{t},\;a_{t}\right)\right],$$
where the action-value function Q(s,a) represents the expected return obtained by taking a certain action *a* in a state *s* based on the policy function *π*. The reward value R(s,a) is the expected value of the sum of immediate rewards for all possible actions at any given time *t*, denoted as $R(s,a)=E\left[R_{t+1}|S_{t}=s,A_{t}=a\right]$, in which s and a represent the state and action at time *t*, respectively.

The objective of the SAC algorithm in reinforcement learning is to maximize the entropy-regularized reward, which is the sum of cumulative reward and policy entropy. In other words, the SAC algorithm incorporates policy entropy in addition to maximizing the cumulative reward in classical reinforcement learning algorithms. The process of finding the optimal policy in the SAC algorithm is represented by Equation (3).
(3)$$\pi^{*}=\arg\underset{\pi }{\max}\underset{\tau \sim \pi }{E}\left[\sum_{t=0}^{\infty }\gamma^{t}\left(\underset{\mathrm{reward}\;}{\underset{︸}{R\left(s_{t},a_{t},s_{t+1}\right)}}+\underset{\mathrm{entropy}\;}{\underset{︸}{\alpha H\left(\pi \left(\cdot |s_{t}\right)\right)}}\right)\right]$$

In Equation (3), α is the entropy regularization coefficient, which adjusts the relative importance of reward and entropy. A higher value of α indicates a larger proportion of entropy, prompting the intelligent agent to explore the environment and employ diverse actions to achieve its objectives. Conversely, a lower value of α implies a reduced emphasis on entropy, causing the intelligent agent to rely more on existing actions to accomplish its objectives. A trajectory τ is a sequence of states and actions, where $\tau =\left(s_{0},a_{0},s_{1},a_{1},\cdots \right)$. γ∈(0,1) is the discount factor. Similarly, compared to classical reinforcement learning algorithms, the SAC algorithm adds entropy rewards to both the value function $V^{\pi }$ and the action-value function $Q^{\pi }$. The specific definitions are shown in Equations (4) and (5),
(4)$$V^{\pi }\left(s\right)=\underset{\tau \sim \pi }{E}\left[\sum_{t=0}^{\infty }\gamma^{t}\left(R\left(s_{t},a_{t},s_{t+1}\right)+\alpha H\left(\pi \left(\cdot |s_{t}\right)\right)\right)|s_{0}=s\right]$$
(5)$$Q^{\pi }\left(s,a\right)=\underset{\tau \sim \pi }{E}\left[\sum_{t=0}^{\infty }\gamma^{t}R\left(s_{t},a_{t},s_{t+1}\right)+\alpha \sum_{t=1}^{\infty }\gamma^{t}H\left(\pi \left(\cdot |s_{t}\right)\right)|s_{0}=s,a_{0}=a\right]$$

The relationship between $V^{\pi }\left(s\right)$ and $Q^{\pi }\left(s\right)$ is as follows:(6)$$V^{\pi }\left(s\right)=\underset{a\sim \pi }{E}\left[Q^{\pi }\left(s,a\right)\right]+\alpha H\left(\pi \left(\cdot |s\right)\right)$$
and the Bellman equation for $Q^{\pi }\left(s\right)$ is
(7)$$\begin{array}{l} Q^{\pi }(s,a)=\underset{\begin{array}{c} s^{\prime }\sim P\\ a^{\prime }\sim \pi \end{array}}{E}\left[R\left(s,a,s^{\prime }\right)+\gamma \left(Q^{\pi }\left(s^{\prime },a^{\prime }\right)+\alpha H\left(\pi \left(\cdot |s^{\prime }\right)\right)\right)\right]\\ \;\;\;=\underset{s^{\prime }\sim P}{E}\left[R\left(s,a,s^{\prime }\right)+\gamma V^{\pi }\left(s^{\prime }\right)\right] \end{array}.$$

Based on the above steps, the policy network can iteratively update its network parameters using the iterative Bellman equation in order to obtain the optimal policy.

#### 3.2.2. Updating Process of SAC Algorithm

This paper utilizes an entropy-weighted SAC algorithm with automatic adjustment of α. The algorithm consists of two *Q* networks (Critic networks) and one Actor network (Policy network). The subsequent sections will present the updating processes of the *Q* networks and the Actor network.

Updating Process of the *Q* Networks

The *Q* networks are updated by sampling experiences $\left(s_{t},a_{t},r_{t},s_{t+1},d\right)$ from the experience replay buffer. The estimation of the state-action value for the *Q* networks is calculated by
(8)$$Q^{\pi }(s_{t},a_{t})\approx r_{t}+\gamma \left(Q^{\pi }\left(s_{t+1},{\tilde{a}}_{t+1}\right)-\alpha \log\pi \left({\tilde{a}}_{t+1}|s_{t+1}\right)\right),\;{\tilde{a}}_{t+1}\sim \pi \left(\cdot |s_{t+1}\right)$$
where ${\tilde{a}}_{t+1}$ is the prediction of the action $a_{t+1}$ by the Actor network. The SAC algorithm employs the mean squared loss function as its loss. Define $D={\left\{\tau_{i}\right\}}_{i=1,\cdots ,N}$ as a set of trajectories, where each trajectory is obtained by letting the agent act in the environment using the policy $\pi_{\theta }$. The loss function for the *Q* networks is defined as follows:(9)$$L\left(\phi_{j},D\right)=\frac{1}{\left|D\right|}\sum_{\left(s_{t},a_{t},r_{t+1},s_{t+1},d\right)\in D}\left[{\left(Q_{\phi_{j}}\left(s_{t},a_{t}\right)-Q_{\mathrm{target}\;}\left(r_{t},s_{t+1},d\right)\right)}^{2}\right],\;j=1,2$$
where |D|=N is the number of trajectories in D.

The SAC algorithm incorporates the clipped double-*Q* technique during the training of the *Q* networks. Consequently, $Q_{target}\left(r_{t},s_{t+1},d\right)$ takes the minimum *Q*-value between the two *Q* approximators. The specific definition is as follows:(10)$$Q_{target}\left(r_{t},s_{t+1},d\right)=r_{t}+\gamma (1-d)\left(\underset{j=1,2}{\min}Q_{\phi_{targ,j}}\left(s_{t+1},{\tilde{a}}_{t+1}\right)-\alpha \log\pi_{\theta }\left({\tilde{a}}_{t+1}|s_{t+1}\right)\right),\;{\tilde{a}}_{t+1}∼\pi_{\theta }\left(\cdot |s_{t+1}\right).$$

2.Updating Process of the Actor Network

The SAC algorithm utilizes a squashed Gaussian policy to select actions, which means that action samples are obtained according to
(11)$${\tilde{a}}_{\theta }\left(s,\xi \right)=\tanh\left(\mu_{\theta }\left(s\right)+\sigma_{\theta }\left(s\right)⊙\xi \right),\;\xi \sim N(0,I),$$
where θ represents the parameters of the Actor network $\pi_{\theta }$, $\mu_{\theta }(s)$ and $\sigma_{\theta }(s)$ respectively correspond to the mean and standard deviation of the action distribution outputted by the Actor network $\pi_{\theta }$, and ξ denotes a random noise that follows a normal distribution. The reparameterization technique is used to optimize the policy in order to rewrite the expectation over actions into an expectation over noise, as follows:(12)$$\underset{a\sim \pi_{\theta }}{E}\left[Q^{\pi_{\theta }}\left(s,a\right)-\alpha \log\pi_{\theta }\left(a|s\right)\right]=\underset{\xi \sim N}{E}\left[Q^{\pi_{\theta }}\left(s,{\tilde{a}}_{\theta }\left(s,\xi \right)\right)-\alpha \log\pi_{\theta }\left({\tilde{a}}_{\theta }\left(s,\xi \right)|s\right)\right].$$

To obtain the policy loss, the final step is to substitute $Q^{\pi_{\theta }}\left(s,{\tilde{a}}_{\theta }\left(s,\xi \right)\right)$ with one of two function approximators. The policy of the Actor network $\pi_{\theta }$ is thus optimized according to
(13)$$L\left(\pi_{\theta },D\right)=\underset{\theta }{\max}\underset{\begin{array}{c} s\sim D\\ \xi \sim N \end{array}}{E}\left[\underset{j=1,2}{\min}Q_{\phi_{j}}\left(s,{\tilde{a}}_{\theta }\left(s,\xi \right)\right)-\alpha \log\pi_{\theta }\left({\tilde{a}}_{\theta }\left(s,\xi \right)|s\right)\right],$$
where $\underset{j=1,2}{\min}Q_{\phi_{j}}\left(s,{\tilde{a}}_{\theta }\left(s,\xi \right)\right)$ is the minimum of the two *Q* approximators.

#### 3.2.3. LSTM Network

During the process of mobile robot path planning, individual states often fail to provide sufficient information to guide the robot in making optimal decisions. As a result, this paper proposes enhancements to the neural network of the SAC algorithm by incorporating LSTM (Long Short-Term Memory) neural networks [35]. By introducing LSTM networks, the algorithm gains the ability to retain and utilize past states alongside the current state, enabling it to make more informed and rational decisions.

Conventional fully connected neural networks are limited in their ability to effectively address time-dependent problems, as their outputs are solely determined by the input at the current time step. However, the Recurrent Neural Network (RNN) provides a solution to this challenge. By incorporating recurrent networks within their architecture, RNNs have demonstrated remarkable efficacy in handling time-dependent problems. Nevertheless, during the training process, RNNs are susceptible to issues such as gradient vanishing or exploding. LSTM networks introduce the gated mechanisms on the foundation of RNNs, allowing for selective information retention and addressing the limitations of traditional RNNs. The LSTM structure comprises three essential gates: the forget gate *f*, the input gate *i*, and the output gate *o*. The forget gate ***f*** determines how much information from the previous time step’s memory cell $C_{t-1}$ should be retained for the current time step’s memory cell $C_{t}$. The input gate ***i*** regulates the amount of current time step’s information to be stored in the candidate state ${\tilde{C}}_{t}$. The output gate ***o*** controls the extent to which information from the current time step’s memory cell $C_{t}$ should be conveyed to the current hidden state $Y_{t}$. The diagram depicting the structure of the LSTM neural networks is illustrated in Figure 2.

In Figure 2, *t* refers to the current time. *X_t_*_−1_, *X_t_*, and *X_t_*_+1_ correspond to the inputs of the previous, current, and next time steps, respectively. Similarly, *Y_t_*_−1_, *Y_t_*, and *Y_t_*_+1_ denote the outputs of the previous, current, and next time steps, respectively. In addition, *C_t_*_−1_, *C_t_*, and *C_t_*_+1_ represent the memory cells at the previous, current, and next time steps, respectively. σ is the sigmoid function, defining output values between 0 and 1. The output values of the function tanh range from −1 to 1.

The workflow of LSTM is as follows. To begin with, the forget gate selectively discards information from the memory cell *C_t_*_−1_ of the previous time step. In order to obtain the forget coefficient, the previous hidden state information *Y_t_*_−1_ and the present input information *X_t_* are both passed through the sigmoid function. The forget gate *f_t_* is obtained by
(14)$$f_{t}=\sigma \left(W_{f}\cdot \left[Y_{t-1},X_{t}\right]+b_{f}\right)$$
where $W_{f}$ and $b_{f}$ represents the weight and bias of the layer network, respectively.

The second step involves generating the information required to update the current memory unit $C_{t}$. This process is divided into two parts. Firstly, the update value $i_{t}$ is generated through the sigmoid layer of the input gate. Secondly, a new candidate value ${\tilde{C}}_{t}$ is generated using the tanh layer. The specific calculations are defined by Equations (15) and (16).
(15)$$i_{t}=\sigma \left(W_{i}\cdot \left[Y_{t-1},X_{t}\right]+b_{i}\right)$$
(16)$${\tilde{C}}_{t}=\tanh\left(W_{C}\cdot \left[Y_{t-1},X_{t}\right]+b_{C}\right)$$

The current memory unit $C_{t}$ at the current time step is defined by Equation (17). It is computed as the sum of two products: the product of the memory unit at the previous time step *C_t_*_−1_ and the forget gate control signal *f_t_*, and the product of the input gate value $i_{t}$ and the candidate value ${\tilde{C}}_{t}$.
(17)$$C_{t}=f_{t}\cdot C_{t-1}+i_{t}\cdot {\tilde{C}}_{t}$$

The final step involves determining the output of the LSTM model. Firstly, the control signal $o_{t}$ of the output gate is obtained using the sigmoid function. Secondly, the current memory unit $C_{t}$ at the current time step is scaled by applying the tanh function. Multiplying these two values yields the current output value $Y_{t}$. The specific calculations are defined by Equations (18) and (19).
(18)$$o_{t}=\sigma \left(W_{o}\cdot \left[Y_{t-1},X_{t}\right]+b_{o}\right)$$
(19)$$Y_{t}=o_{t}\cdot \tanh\left(C_{t}\right)$$

Figure 3 illustrates the network structure of the SAC algorithm after incorporating LSTM. This diagram represents the final network architecture of the SAC-LSTM algorithm, as subsequent enhancements do not pertain to the network structure.

The network architecture of the SAC-LSTM algorithm consists of two components: the Actor network and the Critic network. The Actor network begins with a fully connected layer, FC1, comprising 256 neurons. It takes the state feature vector ***s*** as input and utilizes the ReLU activation function to enhance feature extraction. The subsequent layer is a memory-capable LSTM layer with 256 neurons, enabling the algorithm to incorporate historical and current states for improved decision making. Following the LSTM layer, there is a fully connected layer, FC2, with 256 neurons that applies the ReLU activation function to process the LSTM layer’s outputs. The final layer is another fully connected layer with four neurons, producing the mean μ and standard deviation σ, which are used to resample from a Gaussian distribution N=(μ,σ). The resulting action ***a*** is obtained by applying the tanh activation function.

The Critic network receives both the state vector ***s*** and the action vector ***a*** as inputs. The FC1 and LSTM layers mirror those of the Actor network. The FC2 layer, a fully connected layer with 16 neurons, extracts the features from ***a****,* using the ReLU activation function. The FC3 layer takes the concatenated features from the LSTM and FC2 layers as input, comprising 272 neurons and utilizing the ReLU activation function. Finally, the output layer consists of a single neuron, which outputs the *Q*-value for updating the Actor network.

#### 3.2.4. Burn-In Mechanism

Updating LSTM networks requires a series of consecutive sequential samples. However, the high correlation among consecutive sequence samples can lead to increased variance in parameter updates. Therefore, the currently predominant approach is to employ the random order update method, as utilized in DRQN. This approach involves selecting a complete episode of experiences from the experience replay buffer and randomly choosing a fixed-length continuous sequence from that episode to train the algorithm. Prior to each training iteration, the LSTM’s hidden state $h_{\mathrm{int}}$ is initialized to zero. Figure 4 provides a schematic representation of the random order update method, wherein the orange circles represent individual experience tuples $\left(s_{t},a_{t},r_{t+1},s_{t+1},d\right)$, the black boxes represent sequence experiences of length L, and DRL refers to a deep reinforcement learning algorithm incorporating an LSTM network. This updating method offers the advantage of simplicity and low complexity. However, resetting the LSTM network’s hidden state to zero before training can lead to impaired memory within the LSTM network, subsequently impacting the algorithm’s performance.

Therefore, the burn-in mechanism utilized in the R2D2 algorithm [36] is introduced in this paper. The burn-in mechanism serves as a warm-up mechanism, initializing the LSTM network’s hidden state $h_{\mathrm{int}}$ with a portion of historical data prior to training. Figure 5 depicts the conceptual diagram illustrating the application of the burn-in period in deep reinforcement learning algorithms. The black box represents a sequence of data, while the green circles represent the burn-in data comprising $l_{b}$ items. The red circles represent the training data consisting of $l_{t}$ items. When the sequence data are sampled by the DRL algorithm, the first $l_{b}$ items are used to update the LSTM’s hidden state $h_{t}$ within the DRL framework. Subsequently, the remaining $l_{t}$ items are utilized to train the DRL algorithm. By incorporating the burn-in mechanism, the hidden state of the LSTM network is updated before training, thereby circumventing the issue of impaired memory capacity caused by zeroing the LSTM network’s hidden state prior to training. Consequently, this integration enhances the performance of the algorithm.

This study treats a fixed-length sequence $\left\{\left(s_{t},a_{t},r_{t+1},s_{t+1},d),\ldots ,(s_{t+L},a_{t+L},r_{t+1+L},s_{t+1+L},d\right)\right\}$ of L as a single experience. As the experience tuples during the burn-in period are not utilized for network training, an approach is adopted to prevent experience wastage. Specifically, the storage format of experiences is designed in such a way that there is a duplication of half the experience tuple between two adjacent experiences. Figure 6 provides a visual representation of the specific storage format, with each circle representing an individual experience tuple $\left(s_{t},a_{t},r_{t+1},s_{t+1},d\right)$.

#### 3.2.5. Prioritized Experience Replay

During the exploration of the environment, the agent accumulates training data, which is then stored in an experience replay buffer in the form of experience tuples. However, these samples tend to be sparse, and there exists a strong correlation among consecutively collected samples. Hence, in the context of deep reinforcement learning training, it is essential to employ random sampling of the samples within the experience replay buffer. This approach serves to enhance the efficiency of data utilization, disrupt the inter-sample correlation, and effectively mitigate the occurrence of overfitting in neural networks.

The SAC algorithm employs random sampling during training, which enhances the efficiency of data utilization and mitigates neural network overfitting to a certain extent. However, the assumption of equal importance for all experience samples in random sampling is not aligned with reality. In practice, different experience samples possess varying levels of significance. For instance, experiences with high success rates or frequent failures hold greater value for the algorithm’s learning process, as they can expedite convergence. Consequently, this paper integrates the concept of prioritized experience replay with the SAC algorithm. This integration allows the agent to discern the importance of experience samples and prioritize frequent sampling of high-value samples, thus accelerating the convergence speed of the algorithm.

In the context of reinforcement learning, TD-error is commonly utilized to quantify the importance of samples, with a higher TD-error value indicating a greater degree of significance for the respective sample. By prioritizing the learning from samples with larger TD-errors, the algorithm can expedite the rate of learning. Specifically, in the prioritized experience replay DQN algorithm, the TD-error of each experience tuple $\left(s_{t},a_{t},r_{t},s_{t+1},d\right)$ is defined as the error $\delta_{t}$ between the current *Q*-value and the target Q-value, as illustrated in Equation (20), in which $Q_{target}$ and *Q* respectively represent the target Q-network and the current Q-network.
(20)$$\delta_{t}=r\left(s_{t},a_{t}\right)+\gamma Q_{target}\left(s_{t+1},{\tilde{a}}_{t+1}\right)-Q\left(s_{t},a_{t}\right)$$

Unlike the DQN algorithm, the SAC algorithm incorporates two Q-networks. This paper defines the absolute value $\left|\delta_{t}\right|$ of the TD-error for an experience tuple as the average of the absolute TD-errors from the two Q-networks, as precisely specified in Equation (21).
(21)$$|\delta_{t}|=\frac{1}{2}\sum_{j=1}^{2}\left|Q_{\phi_{j}}\left(s_{t},a_{t}\right)-Q_{\mathrm{target}\;}\left(r_{t},s_{t+1},d\right)\right|$$

As the LSTM network is introduced, the storage format of experiences has transformed into the form depicted in Figure 6. Moreover, in conjunction with the burn-in mechanism, the TD-error of an experience sample $\left\{\left(s_{t},a_{t},r_{t+1},s_{t+1},d),\ldots ,(s_{t+L},a_{t+L},r_{t+1+L},s_{t+1+L},d\right)\right\}$ is defined as the absolute TD-error of the subsequent $l_{t}$-term experience tuples. This specific definition is presented in Equation (22).
(22)$$\delta =\frac{1}{l_{t}}\sum_{t=l_{b}+1}^{l_{b}+l_{t}}\left|\delta \left(s_{t},a_{t},r_{t},s_{t+1},d\right)\right|,$$

The sampling probability [37] for an experience sample is expressed by
(23)$$P(i)=\frac{p_{i}^{\alpha_{p}}}{\sum_{k}p_{k}^{\alpha_{p}}},\alpha_{p}\in [0,1].$$

In Equation (23), the exponent $\alpha_{p}$ serves as the coefficient for regulating prioritization. When $\alpha_{p}$= 0, the sampling method reverts to uniform sampling. $p_{i}$ > 0 is the priority of transition *i* based on TD-error, employing a proportional prioritization approach defined in Equation (24).
(24)$$p_{i}=\left|\delta_{i}\right|+\varepsilon$$

In Equation (24), ε is typically a small positive value that ensures the inclusion of experience samples with a TD-error of 0. However, prioritizing samples with larger TD-error values can disrupt the probability distribution of training samples. This approach may introduce bias and potentially hinder the convergence of the neural network. Therefore, it is necessary to incorporate importance sampling to adjust the learning rate of the samples. The specific definition is provided in Equation (25).
(25)$$w_{i}={\left(\frac{1}{N}\cdot \frac{1}{P\left(i\right)}\right)}^{\beta }$$

In Equation (25), *N* represents the capacity of the experience replay buffer, while β is a hyperparameter for error correction that ranges between 0 and 1. By distinguishing the importance of experiences using the aforementioned approach, it enhances the learning efficiency of the algorithm.

#### 3.2.6. SAC-LSTM Algorithm Workflow

Based on the SAC algorithm, this paper introduces the SAC-LSTM algorithm by incorporating LSTM neural networks, burn-in, and prioritized experience replay. The workflow of the algorithm and its corresponding pseudocode are presented in Figure 7 and Algorithm 1, respectively. The agent interacts with the environment, generating experience tuples (st,at,rt,st+1,d) that are subsequently stored in the experience replay buffer. By employing prioritized experience replay, the algorithm effectively samples experiences. The integration of the burn-in mechanism, as indicated by the green circle in Figure 7, enhances the efficiency of the training process.
**Algorithm 1**: SAC-LSTM algorithm pseudocode1: randomly initialize the parameter θ of the actor network and the parameters ϕ1,ϕ2 of the Critic network, clear the experience replay buffer D2: initialize the target networks ϕtarg,1←ϕ1,ϕtarg,2←ϕ2, set the length of the burn-in data and the lengths lb,lt of the training data, set the capacity N of the experience replay buffer3: **for** episode = 1 to M **do**4: initialize the observation s1 and the hidden state h15: **for** t = 1 to T **do**6:  obtain the observation st and select an action at using the current policy network7:  perform action at,obtain next observation st+1, receive reward rt     determine whether the current state is a terminal state through the signal d8:  store (st,at,rt,st+1,d) into D9: **end for**10: assign priority Pt=maxi<tPi to experience [(st,at,rt,st+1,d),…,(st+L,at+L,rt+L,st+1+L,d)]11: sample N experiences from D based on their priority P(j) and reset the hidden state to zero12: Calculate the importance weight wi for each experience sample13: Scan the previous lb experiences for each sample and obtain the initial hidden state ht14: Calculate the target *Q*-function values y1i,⋯,ylti using the last lt experiences:yti=rti+γ(1−d)(minj=1,2Qϕtarg,j(st+1i,a˜t+1)−αlogπθ(a˜t+1|st+1i)), i=1,2, a˜t+1~πθ(⋅|st+1i)15: Update the priority pi←|δi| based on the TD-error16: Update Q network using gradient descent method with the following formula:    ∇ϕj1|N⋅lt|∑iwi∑t[(Qϕj(sti,ati)−yti)2], i=1,217: Update policy network using gradient descent method with the following formula:     ∇θ1|N⋅lt|∑i∑t(minj=1,2Qϕj(sti,a˜t)−αlogπθ(a˜t|sti)), i=1,2, a˜t+1~πθ(⋅∣sti)18: Update target network ϕtarg,j←ρϕj+(1−ρ)ϕtarg, jj=1,219: End for

### 3.3. Motion Model of Mobile Robot

Turtlebot3 is a cost-effective and open-source mobile robot that offers a simple yet powerful design. It boasts high expandability and the ability to easily integrate additional sensors as needed. Consequently, this paper selects Turtlebot3 as the platform for algorithm deployment. Operating on a differential drive system, this robot can execute turns, maintain a constant velocity, and rotate in place by controlling the differential motion of its two rear wheels. The model structure is depicted in Figure 8.

The pose of a mobile robot in the world coordinate system is represented by the coordinates ${\left[x,y,\psi \right]}^{T}$. Here, *x* and *y* correspond to the central point of the robot in the *X-* and *Y*-axes of the world coordinate system. The parameter ψ denotes the orientation of the robot, indicating the angle between its forward direction (aligned with the linear velocity v) and the *X*-axis of the world coordinate system. The distance between the two drive wheels of the robot is denoted as *W*. Assuming the left and right wheel velocities are $v_{L}$ and $v_{R}$, respectively, the linear velocity v of the mobile robot can be determined using the following equation:(26)$$v=\frac{v_{R}+v_{L}}{2}.$$

The lateral angular velocity ω of the mobile robot is given by
(27)$$\omega =\frac{v_{R}-v_{L}}{W}.$$

The instantaneous radius *R* of the mobile robot during motion is given by
(28)$$R=\frac{v}{\omega }=\frac{W(v_{R}+v_{l})}{2(v_{R}-v_{l})}.$$

The motion equation in global coordinates is
(29)$$\left[\begin{array}{c} \dot{x}\\ \dot{y}\\ \dot{\psi } \end{array}\right]=\left[\begin{array}{cc} \cos\psi & 0\\ \sin\psi & 0\\ 0 & 1 \end{array}\right]\left[\begin{array}{c} v\\ \omega \end{array}\right]$$

### 3.4. Path Planning Algorithm

#### 3.4.1. State Space and Action Space

The state represents the agent’s perception of the environment and serves as the foundation for action selection. Designing a well-defined state space is crucial for the agent to learn an optimal strategy. In the context of path planning tasks performed by a mobile robot, the robot relies on sensor input to perceive the surrounding environment. By leveraging this information, the algorithm generates a collision-free path from the initial position to the goal. Consequently, the chosen state space in this paper revolves around two key aspects: sensor perception information and the position of the goal.

For our experimental setup, we employed the Turtlebot3 mobile robot equipped with a laser scanner for environment perception. The laser scanner provides a detection range of [0°, 360°], resulting in a 360-dimensional data representation. To mitigate the issue of high dimensionality without compromising the effectiveness of capturing environmental information, we adjusted the detection range of the laser scanner to [−90°, 90°] and limited the detection distance to the range of [0.1 m, 3.5 m]. Specifically, the laser scanner’s scan data is represented by 20 dimensions, as illustrated in Figure 9.

In order to expedite the attainment of the designated target point, it is imperative to provide guidance to the robot regarding its trajectory. Thus, this paper adopts the utilization of the heading angular deviation $Rad_{diff}^{t}$ between the frontal orientation of the mobile robot and the target point and the distance between the robot and the target point as the fundamental states for this purpose. By employing the mobile robot’s odometry, the coordinates $\left(x_{robot}^{t},y_{robot}^{t}\right)$ of the robot can be determined, along with the coordinates $\left(x_{goal},y_{goal}\right)$ of the target point as shown in Figure 10. Consequently, the distance $dis_{t}$ between the robot and the target point can be computed.
(30)$$dis_{t}=\sqrt{{\left(x_{goal}-x_{robot}^{t}\right)}^{2}+{\left(y_{goal}-y_{robot}^{t}\right)}^{2}}$$

The heading angle $yaw_{t}$ can be obtained from the odometer, but the quadratic data obtained cannot be used directly, and have to be converted into a Eulerian angle first.
(31)$$yaw_{t}=euler\_from\_quaternion\left(orientation\right)$$

The angle $Rad_{goal}^{t}$ between the mobile robot and the target point can be obtained by the inverse tangent function.
(32)$$Rad_{goal}^{t}=\arctan\left(x_{goal}-x_{robot}^{t},y_{goal}-y_{robot}^{t}\right)$$

The heading angular deviation is the difference between the angle $Rad_{goal}^{t}$ and the orientation angle φ.
(33)$$Rad_{diff}^{t}=Rad_{goal}^{t}-\varphi$$

In this paper, we enhance the state representation of a mobile robot by incorporating the linear velocity and angular velocity from the previous time step. This addition allows for a correspondence between the rewards provided by the environment and the actions performed by the mobile robot. Ultimately, the state space of the mobile robot is defined as a 20-dimensional vector comprising radar detection data, heading angular deviation, linear velocity, and angular velocity from the previous time step, as well as the distance between the target point and the mobile robot.
(34)$$S_{t}=[scan_{t},\nu_{t-1},\omega_{t-1},Rad_{diff}^{t},dis_{t}]$$

Many previous deep reinforcement learning algorithms have used a discretized action space in which the linear and angular velocities of mobile robots were divided into several orders of magnitude. Although this approach is relatively simple, it ignores the fact that mobile robots output continuous actions. To make the simulation more realistic, this paper presents a network model that produces continuous angular and linear velocities. The linear velocity has a range of 0 to 0.22 m/s, while the angular velocity ranges from −2 to 2 rad/s, where positive angular velocity indicates clockwise rotation and negative angular velocity indicates counterclockwise rotation. The proposed approach is intended to bring the simulation closer to real-world scenarios.

#### 3.4.2. Reword Function

The reward function plays a critical role in the success of deep reinforcement learning algorithms as it serves as the benchmark for evaluating agent performance. Similar to constraints used in traditional path planning tasks, the reward function guides agents by indicating which actions to avoid and which ones to pursue given the current state. In this paper, the proposed reward function is composed of two distinct components whose sum constitutes the overall reward:(35)$$R_{total}=R_{a}+R_{b}$$

The specific expressions of $R_{a}$ and $R_{b}$ are shown in Equations (36) and (37). $d_{o}$ represents the distance threshold for reaching the target point, while $d_{\min}$ represents the threshold for avoiding collisions with obstacle. $d_{\max}$ is the safe distance threshold from the obstacle. Furthermore, $d_{t}$ denotes the current distance between the moving robot and the target point, and $d_{t-1}$ represents the previous time’s distance. Additionally, $d_{s\min}^{t}$ is the minimum distance recorded from the radar scan at the current time. The reward coefficients are indicated as $\eta_{1}$, $\eta_{2}$, and $\eta_{3}$.
(36)$$R_{a}=\{\begin{array}{l} r_{a}\;\;\;\;\;\;\;\;\;if\left(d_{t}<d_{o}\right)\\ r_{c\;}\;\;\;\;\;\;\;\;\;if\left(d_{s\min}^{t}<d_{\min}\right)\\ \eta_{1}\left(d_{t-1}-d_{t}\right)+\eta_{2}\left(\pi -\left|\varphi \right|\right)\;otherwise \end{array}$$
(37)$$R_{b}=\left\{ \begin{array}{l} \eta_{3}\left(d_{s\min}^{t}-d_{\max}\right)if\left(d_{\max}>d_{s\min}^{t}>d_{\min}\right)\\ 0\;\;\;\;\;\;\;\;\;otherwise \end{array}\right.$$

This paper proposes a design approach where mobile robots reaching a designated target point or encountering obstacles are associated with sparse rewards, while dense rewards are assigned for other scenarios. This particular design methodology ensures algorithmic stability throughout the training process.

The sparse reward setting is relatively straightforward, involving the assignment of an immediate reward to the mobile robot upon reaching the target point or colliding with an obstacle. When the distance between the mobile robot and the target point is below a threshold value $d_{o}$, it is considered to have successfully reached the target point, resulting in a positive reward $r_{a}$. Conversely, if the distance between the mobile robot and the obstacle is below a threshold value $d_{\min}$, a collision is assumed, leading to a negative reward $r_{c}$.

The dense rewards comprise the distance reward, orientation angle reward, and safety reward. The distance reward, denoted as $\eta_{1}\left(d_{t-1}-d_{t}\right)$, is contingent upon the mobile robot’s velocity while moving towards the target point, with higher rewards for faster velocities and lower rewards for slower velocities. The orientation angle reward, denoted as $\eta_{2}\left(\pi -\left|\varphi \right|\right)$, is determined by the orientation angle φ, where the reward is maximized at 0 when the mobile robot is directly facing the target point, and minimized at π when the mobile robot is facing away from the target point. The sum $\eta_{1}\left(d_{t-1}-d_{t}\right)+\eta_{2}\left(\pi -\left|\varphi \right|\right)$ of the distance reward and orientation angle reward aims to expedite the mobile robot’s arrival at the target point. Additionally, the safety reward, represented by $\eta_{3}\left(d_{s\min}^{t}-d_{\max}\right)$, imposes a negative reward when the distance between the mobile robot and the obstacle is below the designated threshold $d_{\max}$. Moreover, the negative reward increases as the distance decreases. The purpose of the safety reward is to maintain a safe distance from obstacles during the path planning process.

The pseudocode for the reward function is presented in Algorithm 2.
**Algorithm 2**: Reward function pseudocode**Input**: initialized reward coefficients η1, η2, η3, thresholds do,dmax, action at**Output**: reward r dmin1: **for** each step of reward **do**2: performs action at3: calculate dense reward r=η1(dt−1−dt)+η2(π−|φ|) 4: **if**
dmax>dsmint>dmin **then**5:  r+=η3(dsmint−dmax)6: **else if**
dt<do **then**7:   r+=ra8:  **else if**
dsmint<dmin **then**9:    r+=rc10:  **end**11: **end**12: return reward r13: **end for**

#### 3.4.3. Path Planning Algorithm Workflow

The path planning algorithm, based on deep reinforcement learning, essentially involves learning the optimal strategy for the mobile robot to reach the target point quickly through interactive exploration with the environment. The algorithm pseudocode for path planning, utilizing SAC-LSTM, is depicted in Algorithm 3.
**Algorithm 3:** Path planning algorithm pseudocode1: initialize system parameters2: introduce LSTM network to optimize the network architecture3: **for**
*N*_t_ < *N*_tmax_ **do**4: initialize the robot’s position5: get environmental information6: perform actions through the policy network7: obtain new state and reward8: store experiences in the experience buffer9: sample experiences based on their priority10: train the network using the burn-in mechanism11: **if** the robot reaches the target point **then**12:  generate a new target point13:  go to step 514: **else if** the terminal state is not met or *N*_s_ < *N*_smax_
**then**15:   go to step 516:  **else**17:   go to step 318:  **end**19: **end**
20: **end for**

At the beginning of each episode, the mobile robot is positioned at a predefined starting point and subsequently navigates towards a randomly generated target destination. Throughout this process, the mobile robot relies on a laser radar to perceive its surrounding environment and utilizes the state information as input to the policy network of the SAC-LSTM algorithm, producing a corresponding robot action. Following interaction with the environment, the mobile robot transitions to the next state and receives a reward based on a predefined reward function. The acquired experiences are stored in the experience replay buffer, and significant experiences are extracted using a prioritized experience replay mechanism. The SAC-LSTM algorithm is trained using the burn-in strategy. Upon reaching the target point, a new target point is randomly generated in the environment, and the robot moves from its current location towards the newly selected target. The training episode terminates only when the robot collides with an obstacle or when the current step count *N*_s_ reaches the designated maximum value *N*_smax_. The training process concludes when the number of training episodes *N*_t_ surpasses the predetermined maximum value *N*_tmax_.

## 4. Experiments and Discussion

### 4.1. Simulation Platform

The experimental software environment for this paper included Ubuntu 18.04, CUDA 10.1, Pytorch 3.7, and ROS Kinetic. The hardware utilization comprised an AMD R7-5800H CPU and a GeForce GTX 3060 GPU with 6G of memory. The experiment employed the Turtlebot3 robot based on ROS, which obtains the environmental information around it with the help of laser radar. The 3D model of the mobile robot was loaded into the ‘empty.world’ in ROS, as depicted in Figure 11.

This paper utilizes Gazebo9 software to set up experimental environments. Three experimental environments, depicted in Figure 12, were built for testing the algorithm’s effectiveness in mobile robot path planning in indoor environments. These environments are an obstacle-free environment, a static obstacle environment, and a dynamic obstacle environment; all three are square areas measuring 8 m in length. The obstacle-free environment has only four surrounding walls. The static obstacle environment features twelve stationary obstacles, consisting of four cylinders with a diameter of 0.3 m and a height of 0.5 m, four small cubes measuring 0.5 m in edge-length, and four large cubes measuring 0.8 m in edge-length, which are added to the obstacle-free environment. In contrast, the dynamic obstacle environment features moving obstacles. The four cylinders rotate counterclockwise at a speed of 0.5 rad/s around the origin of the coordinate system (the center of the environment), as illustrated by the smaller white dashed circles in Figure 12. Concurrently, four large cubes move clockwise at the same speed, following the trajectory shown by the larger white dashed circles with arrows in Figure 12. Meanwhile, the four small cubes within this environment remain stationary.

Deep reinforcement learning involves agents gathering training data by interacting with their real-world environments. In a 3D experimental environment created by Gazebo, data acquisition can become a time-consuming and computationally expensive process. Consequently, this paper employs a time acceleration technique to reduce the duration of training. By default, Gazebo’s real-time update rate is 1000, and the max step size value is 0.001. When these settings are multiplied, the resulting ratio between the simulation time and the actual time is 1. However, in order to expedite simulations, the value for max step size is adjusted to 0.005, leading to a fivefold increase in simulation speed.

The specific experimental parameters are shown in Table 1.

### 4.2. Computational Complexity

To quantify the computational complexity of SAC and SAC-LSTM algorithms, the contribution of each component to the overall computational load can be analyzed based on five key factors, as follows:Computational Load of Neural Networks

Due to the larger batch size (512) and experience buffer capacity (20,000), the SAC algorithm requires processing more data in the network training steps, leading to an increased computational load. When it comes to the SAC-LSTM algorithm, the inclusion of LSTM layers introduces additional time-dependency computations. A smaller batch size (32) might reduce the computational load per training iteration, but the burn-in process and LSTM’s time dependencies increase the computational load per batch.

2.Prioritized Experience Replay

Prioritized experience replay requires additional computations to maintain a priority queue and update it after each learning step, increasing the computational load, especially with a larger experience buffer.

3.Learning Parameters

The parameters, a learning rate of 0.001 and a discount factor of 0.99, mainly affect the convergence speed and stability of the algorithm, but have a relatively minor direct impact on computational load.

4.Reward Mechanism

The reward coefficients and thresholds influence the efficiency of learning, but have a limited direct impact on computational load.

5.Optimizer

The Adam optimizer is a computationally efficient optimizer but has a higher computational complexity compared to simpler ones such as SGD (Stochastic Gradient Descent).

Overall, the SAC-LSTM algorithm involves higher per-batch computational loads due to time-dependency processing with LSTM, smaller batch sizes, and burn-in processing. The SAC algorithm, although having larger data volumes per batch, might have a slightly lower per-batch computational load than SAC-LSTM, owing to the absence of complex time-series processing. When running both algorithms on the same computer platform, SAC-LSTM consumes approximately 10% more computation time per batch compared to SAC. This finding aligns closely with the qualitative analysis results mentioned above.

### 4.3. Experimental Results and Analysis

#### 4.3.1. Obstacle-Free Experiment

In this paper, we chose the average reward as the evaluation metric for our algorithm. A higher value confirms better performance of the algorithm. The average reward is calculated from Equation (38). Within each episode, $T=\min\left(N_{s\max},T\right)$ represents the number of steps taken by the agent, and *r* is the reward received for each step.
(38)$$\;\mathrm{reward}\;=\sum_{i=1}^{T}r/T$$

The robot’s starting point is the origin, and the target point is generated at random. During a single episode, once the robot reaches the target point, the environment generates a new target point randomly. As a result, the robot may reach multiple target points within a single episode.

Two reinforcement learning algorithms, SAC and SAC-LSTM, were trained for 1000 episodes in the obstacle-free environment. Their average reward curves are illustrated in Figure 13. During the initial training phase, both algorithms demonstrated a decreasing trend in their average reward. This was attributed to the robot frequently circling in place or moving away from the target point, which was observed within the first 10 episodes. However, after that point, both algorithms exhibited a remarkable increase in their average reward. The SAC-LSTM algorithm reached its reward peak and convergence after 130 episodes, whereas the SAC algorithm achieved the same after 180 episodes. Notably, the average reward of the SAC-LSTM algorithm surpassed that of the SAC algorithm after convergence. This observation suggests that the SAC-LSTM algorithm enabled the robot to reach the target point more frequently during each episode’s path planning process, resulting in better path planning performance. Although both algorithms accomplished the task in the obstacle-free environment, the SAC-LSTM algorithm converged faster and yielded a higher average reward after convergence.

To further evaluate the model’s performance, the SAC and SAC-LSTM algorithms underwent 200 tests in the obstacle-free environment. For each test, the target points were randomly generated, and the results are presented in Table 2. Our evaluation reveals that the path planning success rate for the SAC algorithm was 97%. In contrast, the proposed SAC-LSTM algorithm yielded better performance, achieving a success rate of 100%.

Illustrated in Figure 14 is the motion process of the mobile robot towards the target point in the obstacle-free environment based on the SAC-LSTM algorithm. In the figure, the black dot represents the mobile robot, while the blue sector illustrates the range of the radar scan. The red square indicates the location of the target point. Notably, the mobile robot achieves optimal pathway and accurately reaches the target point from the starting point.

#### 4.3.2. Static Obstacle Experiment

The two algorithms were trained for 1000 episodes in the obstacle environment illustrated in Figure 12b. The plotted average reward curves of the two algorithms are depicted in Figure 15. Notably, the static obstacle environment is significantly more complex than the obstacle-free environment since collision with the surrounding obstacles is more frequent during robot exploration. As expected, greater exploration time was required. The SAC-LSTM algorithm began to converge from the 20th episode. Interestingly, its converging process had a higher average reward than the SAC algorithm, finishing the process at the 380th episode. On the other hand, the SAC algorithm had reduced fluctuations and began to converge from the 30th episode, while the converging process completed at the 500th episode. Impressively, the SAC-LSTM algorithm yielded a faster convergence rate and a higher average reward after convergence than the SAC algorithm.

Two algorithms were tested 200 times in the static obstacle environment, where the target points were randomly generated. The test results are presented in Table 3, indicating that the SAC algorithm’s success rate was only 88.5%, while the proposed SAC-LSTM algorithm achieved a success rate of 95.5%.

Figure 16 illustrates the movement process of a mobile robot based on the SAC-LSTM algorithm, from its starting point to the target point in the static obstacle environment. As shown, the mobile robot avoided the obstacles in the environment and reached the target point via the optimal path.

#### 4.3.3. Dynamic Obstacle Experiment

The dynamic obstacle environment was constructed as shown in Figure 12c to simulate a realistic path planning scenario. The SAC and SAC-LSTM algorithms underwent 1000 episodes of training, with the training results presented in Figure 17. As shown in the figure, the difficulty of path planning increased due to the moving obstacles, resulting in significant fluctuations in the learning curves of both algorithms. The SAC-LSTM algorithm began to converge after the 75th episode when the fluctuations decreased, and it achieved convergence by the 510th episode. The SAC algorithm began to converge after the 95th episode and achieved convergence by the 710th episode. The proposed SAC-LSTM algorithm demonstrated a faster convergence rate than the SAC algorithm, with a slightly higher average reward after convergence.

A total of 200 tests were also conducted using both algorithms in the dynamic obstacle environment, with the target points generated randomly. Table 4 shows the success rates of both algorithms, with the SAC algorithm achieving a success rate of only 78.5%, while the SAC-LSTM algorithm achieved a success rate of 89%.

Figure 18 illustrates the movement process of a mobile robot driven by the SAC-LSTM algorithm, from its starting point to the target point in a dynamic obstacle environment. As shown, the mobile robot was able to avoid the moving obstacles and reach the target point via the optimal path.

As completing path planning tasks in dynamic environments is more challenging and requires higher performance from path planning algorithms, and as dynamic environments better reflect the actual work environment of mobile robots, in order to better test the performance of the two algorithms, an additional set of experiments was conducted on top of the initial experiment measuring success rate, which tested the path length, planning time, and number of times the robot reached the target point. Ten target points were randomly generated in the dynamic obstacle environment, all located near the moving obstacles at the periphery, so that each time the robot moved to a target point, interference from the moving obstacles was encountered, increasing the reliability of the experiment. The specific locations of the target points are illustrated in Figure 19.

The target point numbers in Figure 19 were sorted according to the Euclidean distance between each target point and the starting point of the mobile robot, which is set as the origin. Thirty experimental trials were conducted for each of the target points, with the path length, planning time, and success rates recorded. The path length and planning time were averaged from the successfully completed path planning tasks, and specific test results can be found in Table 5 and Table 6. Among the 10 target points tested, the SAC-LSTM algorithm achieved both shorter average path lengths and planning times, as well as higher success rates, compared to the SAC algorithm. This indicates that the path planning performance of the SAC-LSTM algorithm is superior to that of the SAC algorithm, enabling the mobile robot to reach its designated target point in less time and with a shorter route. Notably, for the 10th target point, the SAC algorithm was unable to guide the mobile robot to its target point based solely on the current state information, whereas the SAC-LSTM algorithm, which incorporates the LSTM network, has memory capability to consider both historical and current states to make better decisions, and thus guided the robot to complete its path planning task.

Based on the results of the three simulation experiments, the trained mobile robot was able to successfully complete the path planning task in all three environments, and the improved SAC-LSTM algorithm demonstrated significant enhancements in both path planning success rate and convergence speed. In particular, in dynamic and complex scenarios, the SAC-LSTM algorithm exhibited shorter planning time, shorter planning paths, and a higher number of instances where the target point was reached.

## 5. Conclusions

This paper presents the SAC-LSTM algorithm and develops a path planning algorithm framework for mobile robots, addressing the limitations of the SAC algorithm in path planning tasks. The proposed algorithm incorporates an LSTM network with memory capability, a burn-in mechanism, and a prioritized experience replay mechanism. By integrating historical and current states, the LSTM network enables more effective path planning decisions. The burn-in mechanism preheats the LSTM network’s hidden state before training, addressing memory depreciation and enhancing the algorithm’s performance. The prioritized experience replay mechanism accelerates algorithm convergence by emphasizing crucial experiences. A motion model for the Turtlebot3 mobile robot was established, and the state space, action space, reward function, and overall process of the SAC-LSTM algorithm were designed. To enhance the realism of the experimental scenarios, three environments were created, including obstacle-free, static obstacle, and dynamic obstacle scenarios. There are multiple stationary and moving obstacles of various sizes and shapes in the dynamic scenarios. The algorithm was subsequently trained and tested in these settings. The experimental results demonstrated that the SAC-LSTM algorithm outperformed the SAC algorithm in convergence speed and path planning success rate across all three scenarios with roughly the same computational cost. Furthermore, in an additional dynamic obstacle experiment, the SAC-LSTM algorithm exhibited shorter planning times, more efficient paths, and an increased number of instances where the target point was reached, indicating superior path planning performance.

Despite these advancements, certain limitations persist within this paper. The experiments relied solely on 2D lidar for environmental data, which may lead to inaccuracies when dealing with irregularly shaped obstacles. Future research could employ multi-sensor fusion to obtain more comprehensive environmental information. Additionally, the paper does not address the several sources of noise and uncertainty present in real-world environments, including sensor noise, environmental fluctuations, and uncertainty in obstacle movement. The algorithm needs sufficient robustness to handle these factors to perform well in practical settings. Moreover, the experiments were conducted exclusively in a simulated environment. To fully assess the effectiveness of the mobile robot in completing path planning tasks, it is essential to transfer the trained model to a real-world scenario.

## Figures and Tables

**Figure 1 sensors-23-09802-f001:**
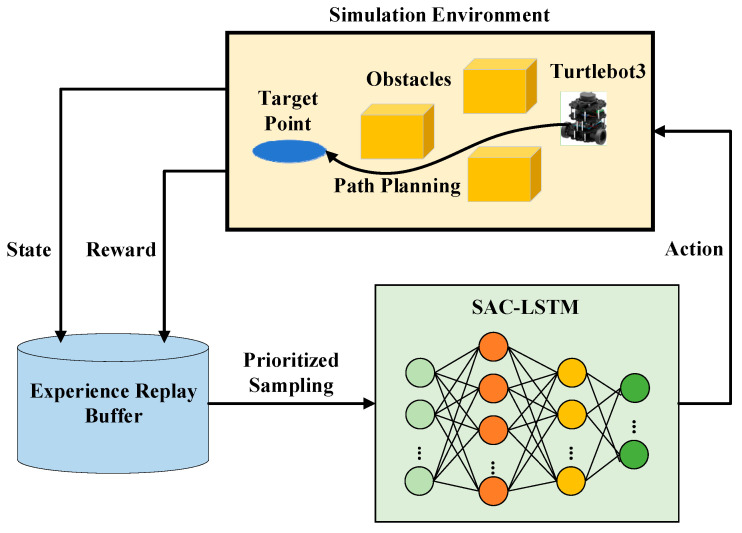
Path planning framework based on SAC-LSTM.

**Figure 2 sensors-23-09802-f002:**
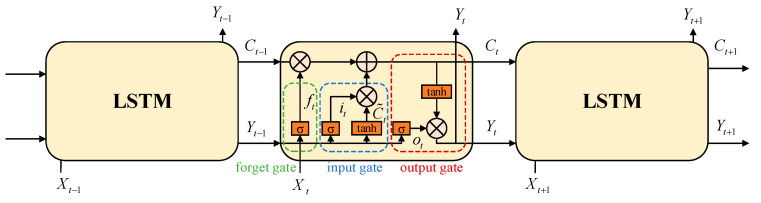
Structure diagram of LSTM neural networks.

**Figure 3 sensors-23-09802-f003:**
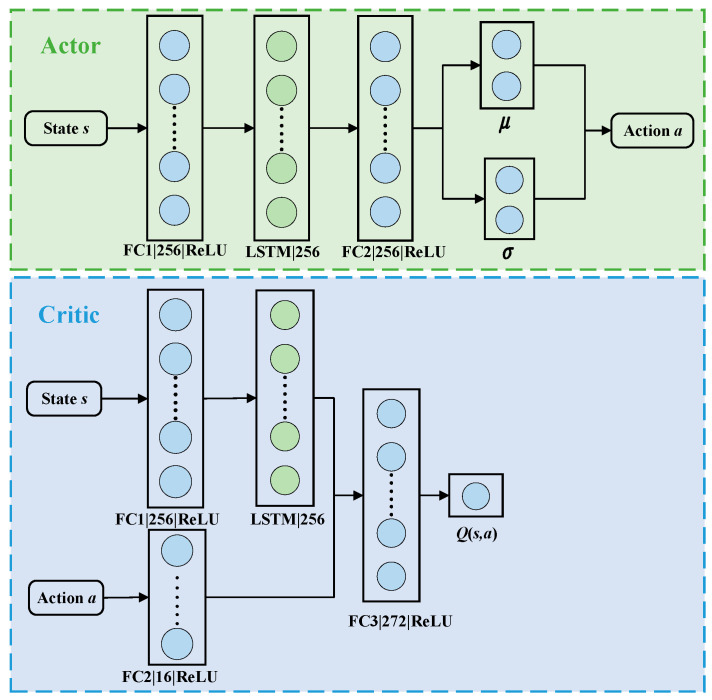
Network structure of SAC-LSTM.

**Figure 4 sensors-23-09802-f004:**
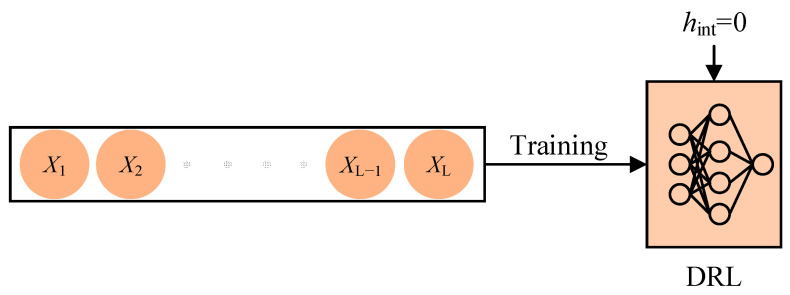
Random order update method.

**Figure 5 sensors-23-09802-f005:**
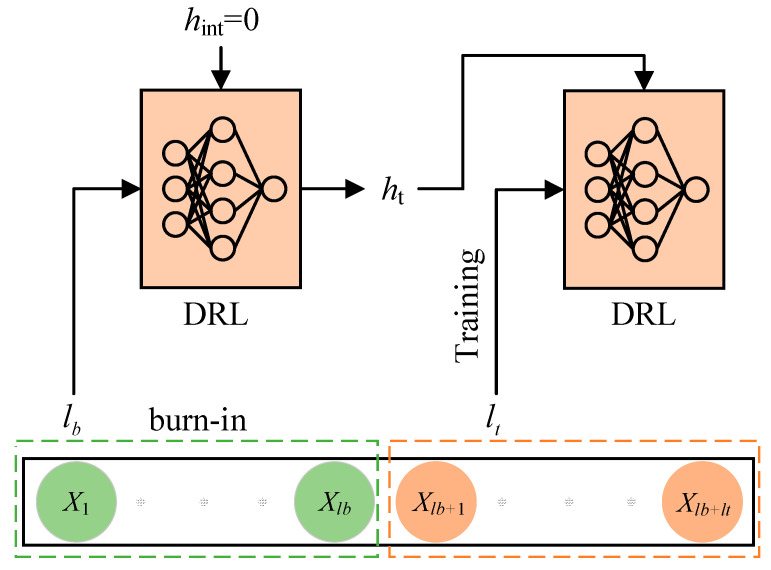
Burn-in period.

**Figure 6 sensors-23-09802-f006:**
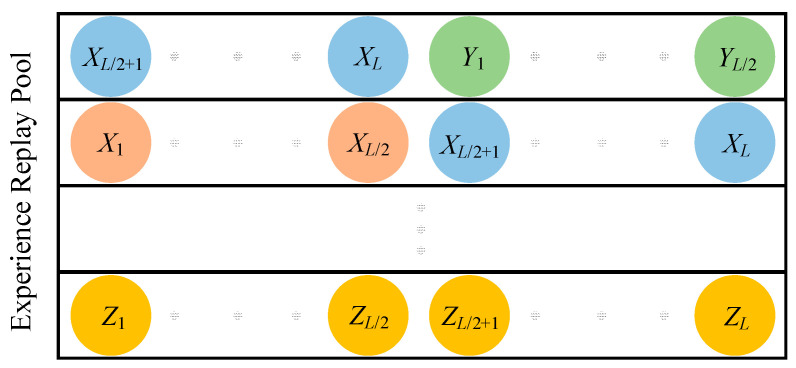
Experience storage format.

**Figure 7 sensors-23-09802-f007:**
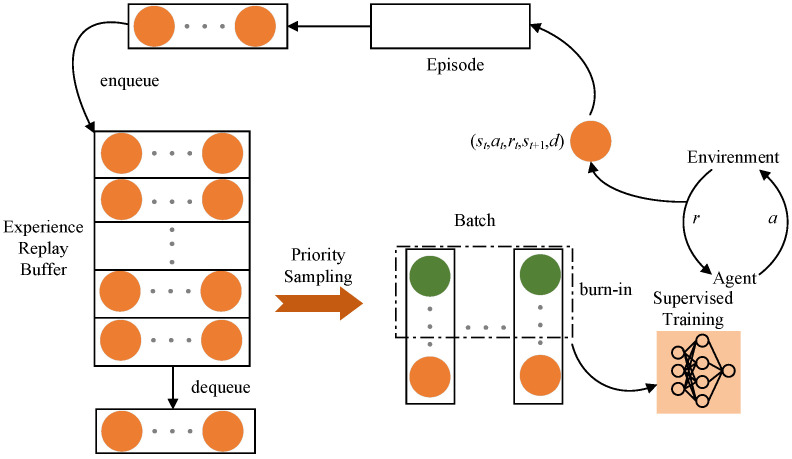
SAC-LSTM algorithm workflow.

**Figure 8 sensors-23-09802-f008:**
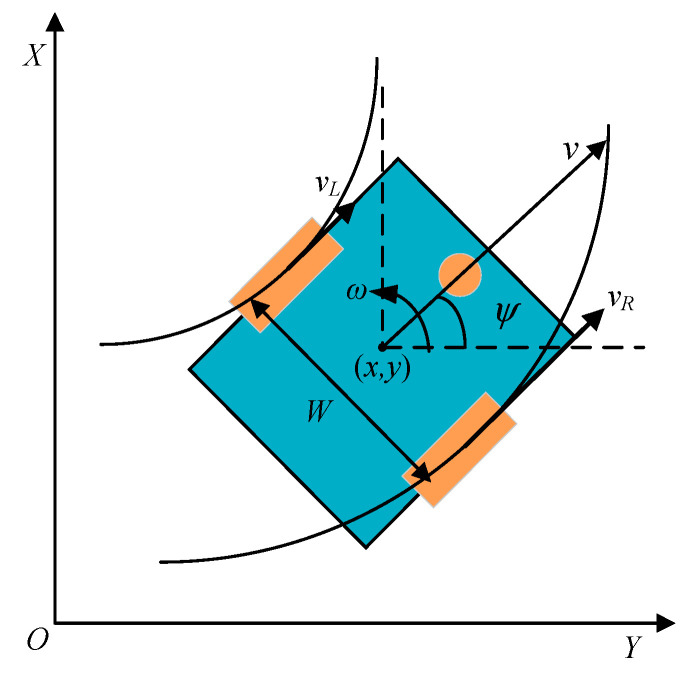
Motion model of the differential drive robot.

**Figure 9 sensors-23-09802-f009:**
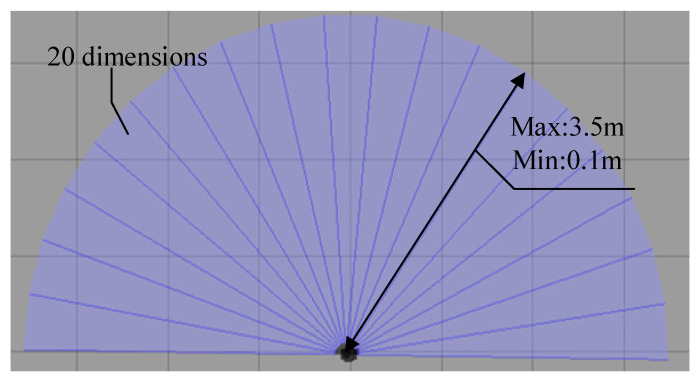
Detection range of the laser sensor on the mobile robot.

**Figure 10 sensors-23-09802-f010:**
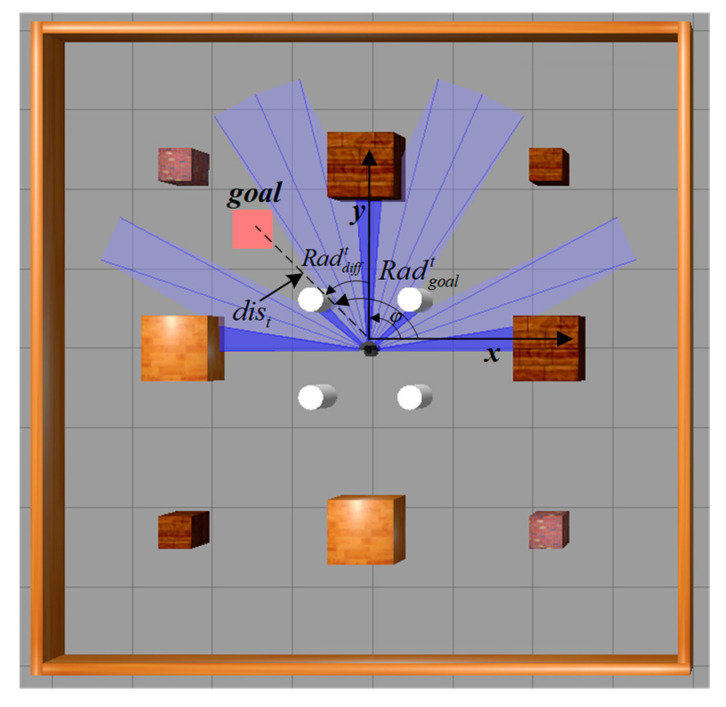
State of the mobile robot.

**Figure 11 sensors-23-09802-f011:**
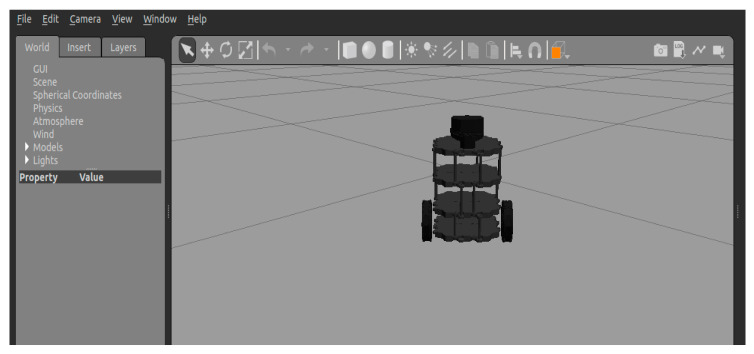
Turtlebot3 3D model.

**Figure 12 sensors-23-09802-f012:**
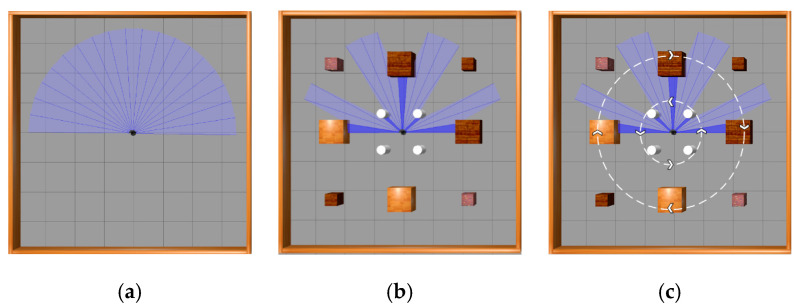
Experimental environments: (**a**) obstacle-free environment; (**b**) static obstacle environment; (**c**) dynamic obstacle environment.

**Figure 13 sensors-23-09802-f013:**
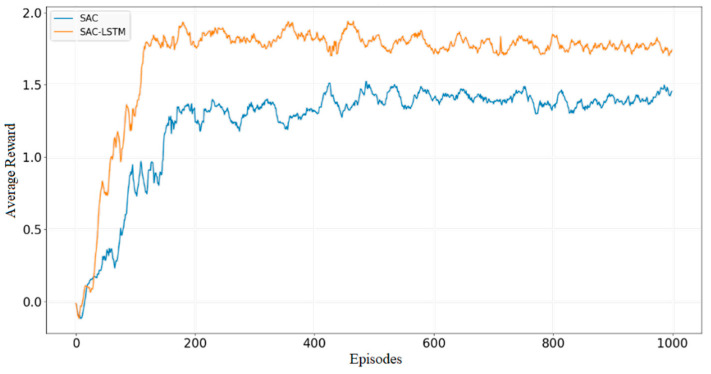
Average reward in the obstacle-free experiment.

**Figure 14 sensors-23-09802-f014:**
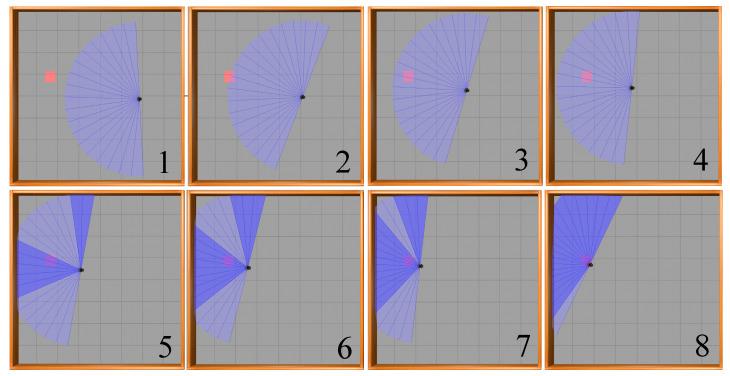
Movement process of the mobile robot in the obstacle-free environment.

**Figure 15 sensors-23-09802-f015:**
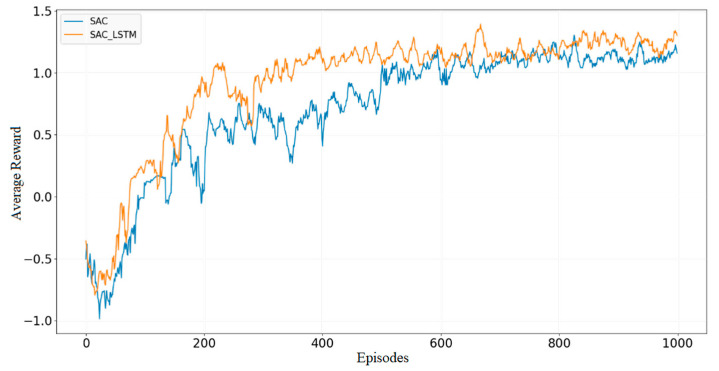
Average reward in static obstacle environment.

**Figure 16 sensors-23-09802-f016:**
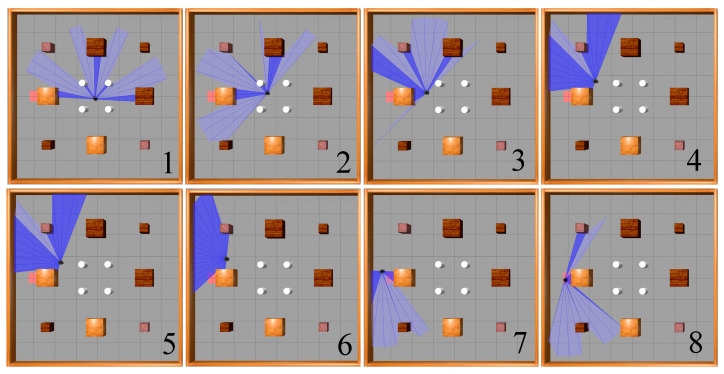
Movement process of the mobile robot in the static obstacle environment.

**Figure 17 sensors-23-09802-f017:**
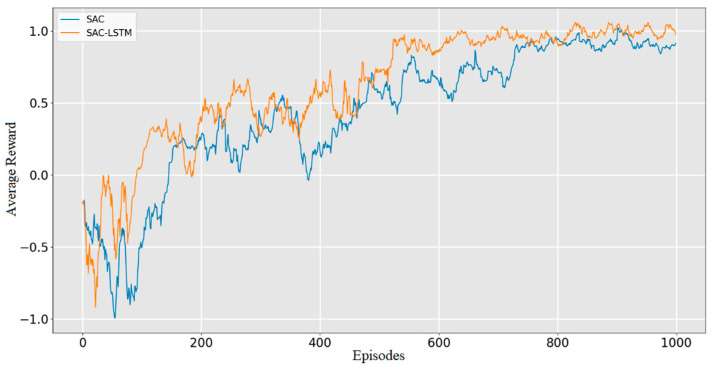
Average reward in the dynamic obstacle environment.

**Figure 18 sensors-23-09802-f018:**
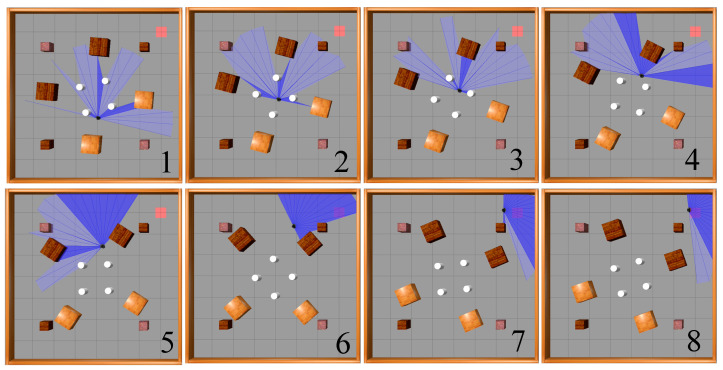
Movement process of the mobile robot in the dynamic obstacle environment.

**Figure 19 sensors-23-09802-f019:**
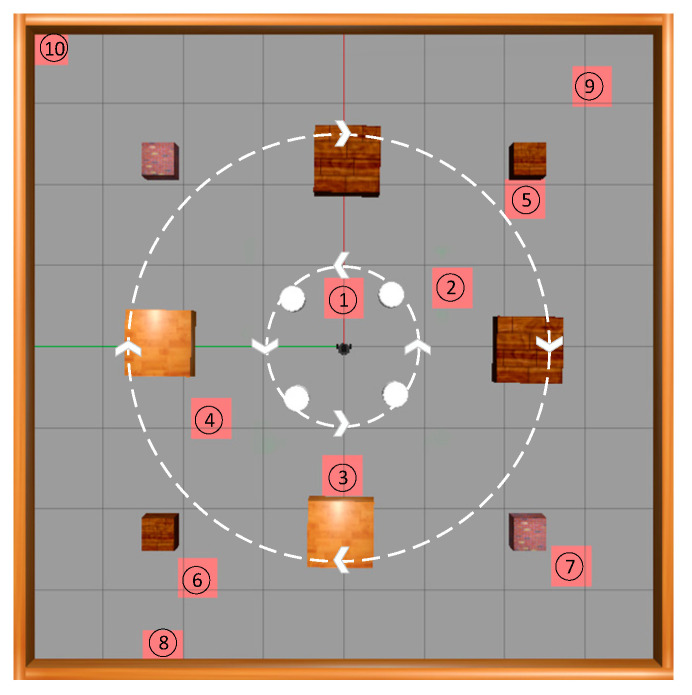
Locations of 10 target points.

**Table 1 sensors-23-09802-t001:** Experimental parameters of mobile robot.

Parameter	Value
Discount factor γ	0.99
Learning rate $l_{r}$	0.001
$\mathrm{Priority}\;\mathrm{regulating}\;\mathrm{coefficient}\;\alpha_{p}$	0.6
β	0.4
Experience buffer capacity (SAC)	20,000
Experience buffer capacity (SAC-LSTM)	5000
Batch size (SAC)	512
Batch size (SAC-LSTM)	32
Burn-in data length $l_{b}$	16
Training length	16
Maximum steps *N*_smax_	500 $l_{t}$
Optimizer	Adam
$\mathrm{Reward}\;\mathrm{coefficient}\;\eta_{1}$	4
$\mathrm{Reward}\;\mathrm{coefficient}\;\eta_{2}$	0.1
$\mathrm{Reward}\;\mathrm{coefficient}\;\eta_{3}$	−3
Reward for reaching the target point $r_{a}$	100
Reward for collision with obstacle $r_{c}$	−50
Distance threshold for reaching the target point $d_{o}$	0.15
Minimum distance threshold from obstacle $d_{\min}$	0.15
Safe distance threshold from obstacle $d_{\max}$.	0.3

**Table 2 sensors-23-09802-t002:** Obstacle-free experiment results.

Algorithm	Success Number	Success Rate
SAC	194	97%
SAC-LSTM	200	100%

**Table 3 sensors-23-09802-t003:** Static obstacle experiment results.

Algorithm	Success Number	Success Rate
SAC	183	88.5%
SAC-LSTM	193	95.5%

**Table 4 sensors-23-09802-t004:** Dynamic obstacle experiment results.

Algorithm	Success Number	Success Rate
SAC	157	78.5%
SAC-LSTM	178	89%

**Table 5 sensors-23-09802-t005:** Test results of the SAC-LSTM algorithm.

Target Point Number	Target Point Location	Path Length (m)	Planning Time (s)	Success Number
1	(0.60, 0.00)	0.64	4.36	30
2	(0.72, −1.35)	1.61	8.89	30
3	(−1.59, 0.02)	1.76	10.47	29
4	(−0.89, 1.65)	1.90	10.98	30
5	(1.82, −2.24)	2.94	16.61	27
6	(−2.86, 1.83)	3.65	20.95	24
7	(−2.69, −2.83)	4.33	24.72	24
8	(−3.74, 2.24)	4.82	27.54	23
9	(3.21, −3.09)	4.85	27.42	25
10	(3.72, 3.68)	5.44	31.22	29

**Table 6 sensors-23-09802-t006:** Test results of the SAC algorithm.

Target Point Number	Target Point Location	Path Length (m)	Planning Time (s)	Success Number
1	(0.60, 0.00)	0.67	5.02	30
2	(0.72, −1.35)	1.75	9.65	30
3	(−1.59, 0.02)	1.89	11.28	24
4	(−0.89, 1.65)	2.03	11.87	27
5	(1.82, −2.24)	3.05	17.68	23
6	(−2.86, 1.83)	3.69	21.83	24
7	(−2.69, −2.83)	4.40	25.08	28
8	(−3.74, 2.24)	4.95	29.63	18
9	(3.21, −3.09)	4.95	27.8	25
10	(3.72, 3.68)	/	/	0

## Data Availability

The data that support the findings of this study are available from the corresponding author upon reasonable request.

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
