# Peer review of "Path Planning of a Mobile Robot for a Dynamic Indoor Environment Based on an SAC-LSTM Algorithm"

_sensors, 2023, doi:10.3390/s23249802_

Round 1

Reviewer 1 Report

Comments and Suggestions for Authors

The SAC-LSTM algorithm is well adapted for this type of research. The objectives of research are well presented and the methodology is matched with these objectives. I recommend to add some technical details on the discussion of the results and also to add some future work possibilities to scale this research.

Comments on the Quality of English Language

The quality of English language could be improved in order to gain more accuracy in the details of discussion and conclusions.

Reviewer 2 Report

Comments and Suggestions for Authors

This paper proposes an improved Soft Actor-Critic Long Short-Term Memory (SAC-LSTM) algorithm for fast path planning of mobile robots in dynamic environments. 

The paper is well written, nevertheless some issues are the following:

1. It is not clear the contribution of the paper. Please make clear statement of the novel contributions. For example, clarify the difference with the following paper:

https://doi.org/10.3390/app12199837

2. Experimental results are required to validate the proposal. This reviewer encourage authors to perform experiments on standard robot platforms.

3. In the abstract authors claim: "...reached the target point more often.". In science subjective terms nust be avoided, please cuantify the results.

4. It is now well known that entropy is not a measure of the disorder of a system (Line 196).

5. Line 249, D=?.

6. Figure 13, 15 and 17 depicts a better performance of  SAC-LSTM over SAC algorithm. However , this reviewer belives a computational comparison of both approaches must be depicted. This might help authors to get sharpen conclussions.

Comments on the Quality of English Language

Minor comments.

Reviewer 3 Report

Comments and Suggestions for Authors

This paper proposes a Soft Actor-Critic Long Short-Term Memory (SAC-LSTM) algorithm for indoor path planning in a dynamic obstacle environment. They combine SAC with LSTM to take advantage of its temporal memory properties, allowing the robot to make decisions based on previous and current observations. The idea is interesting, and the problem being studied is practically important. Simulation results are also presented to support the proposed algorithm. However, the following comments need to be addressed to clarify the algorithm presented in the paper:

* What shortcomings of the SAC algorithm does the paper intend to address? The paper describes what is being presented but doesn't clearly highlight why it is necessary.

* This paper only considers a simplistic model of a mobile robot with a simple kinematic model and no consideration for a dynamic model. How can the proposed model be extended for general mobile robotic systems?

* What is the computational complexity of the proposed algorithm? Since only Simulation results are presented, can the proposed algorithm reliably work in real-time in an actual scenario?

* Continuing to the previous point about computation complexity, how well the system works in rapidly changing dynamic urban environments, e.g., offices, hospitals, shopping malls, etc.

* Is the effect of sensor and environment noise considered in the simulations?

* On use over a long time, how does the SAC-LSTM algorithm adapt to changing environments? Can it learn from long-term environmental changes or obstacle dynamics?

* In addition to LSTM, several metaheuristic-based neural networks have also been applied to control mobile robotic systems, e.g., Tracking control of redundant mobile manipulator: An RNN-based metaheuristic approach. Obstacle avoidance and tracking control of redundant robotic manipulator: An RNN-based metaheuristic approach. Discuss how the proposed LSTM approach compares with these other types of neural network-based control in the introduction of your paper.

Round 2

Reviewer 2 Report

Comments and Suggestions for Authors

Ok, all my comments were directly addressed.

Reviewer 3 Report

Comments and Suggestions for Authors

The authors have addressed the comments. The paper can be accepted.